# New Upgrade to Improve Operation of Conventional Grid-Connected Photovoltaic Systems

Manuel Cáceres [1], Alexis Raúl González Mayans [1], Andrés Firman [1], Luis Vera [1] and Juan de la Casa Higueras [2,*]

1 Group of Renewable Energies (GER), Northeastern National University, Corrientes 3400, Argentina; mcaceres@exa.unne.edu.ar (M.C.); raulgonzalezmayans@exa.unne.edu.ar (A.R.G.M.); afirman@exa.unne.edu.ar (A.F.); luis.horacio.vera@comunidad.unne.edu.ar (L.V.)
2 IDEA Solar Energy Research Group, Center for Advanced Studies in Earth Sciences Energy and Environment, University of Jaén, Las Lagunillas Campus, 23071 Jaén, Spain
* Correspondence: delacasa@ujaen.es; Tel.: +34-615-679152

**Abstract:** The incorporation of distributed generation with photovoltaic systems entails a drawback associated with intermittency in the generation capacity due to variations in the solar resource. In general, this aspect limits the level of penetration that this resource can have without producing an appreciable impact on the quality of the electrical supply. With the intention of reducing its intermittency, this paper presents the characterization of a methodology for maximizing grid-connected PV system operation under low-solar-radiation conditions. A new concept of a hybrid system based on a constant current source and capable of integrating different sources into a conventional grid-connected PV system is presented. Results of an experimental characterization of a low-voltage grid–PV system connection with a DC/DC converter for constant-current source application are shown in zero and non-zero radiation conditions. The results obtained demonstrate that the proposed integration method works efficiently without causing appreciable effects on the parameters that define the quality of the electrical supply. In this way, it is possible to efficiently incorporate another source of energy, taking advantage of the characteristics of the GCPVS without further interventions in the system. It is expected that this topology could help to integrate other generation and/or storage technologies into already existing PV systems, opening a wide field of research in the PV systems area.

**Keywords:** distributed generation; grid integration; hybrid system





## 1. Introduction

Grid-connected PV system (GCPVS) capacity, including both large utility-scale and small distributed systems, accounts for two-thirds of this year's projected increase in global renewable capacity [1]. Nevertheless, a limitation in PV grid penetration appears because of this power generation profile that follows irradiation along the day.

Many works focus on this topic. Ref. [2] showed a comprehensive literature review on associated problems when the intermittent PV is connected, highlighting the voltage and frequency fluctuations; Ref. [3] studied voltage variations in low-voltage distribution networks due to rapid changes in photovoltaic power generation; Ref. [4] focuses on the significance of peak demand reduction in optimizing grid-connected PV battery systems, aiming to achieve a flatter profile; and ref. [5] studied the difficulty in achieving frequency stability of PV systems.

To avoid these inconveniences, many solutions have been proposed: Ref. [6] presents a comparison of several current grid codes for integrating PV systems and explores future grid code amendments for maintaining power system secure operation; in [7–9], they investigate the interaction between PV and other generation systems connected in an output-common DC point. Also, they study models and test different DC-DC topologies; Ref. [10] talks about the modeling and simulation of a hybrid grid-connected system,

including renewable and storage sources. They propose eliminating the PV converter and designing a grid-connected hybrid system, and ref. [11] proposes a voltage regulation methodology based on active-reactive power management through PV inverters. Ref. [12] finds that grid-forming control inverters may improve the frequency profile and terminal voltage during disturbances, and refs. [13,14] propose to reduce the uncertainty of variable renewable production with the forecast of solar energy. From an electrical power system point of view, these solutions are generally used in large power systems, on a macro scale, without considering distributed generation power at the residential–commercial user level.

Additionally, low power GCPVS without storage operates only when solar radiation hits the PV module surface, and the rest of the time both modules and inverters rest. Compared to another generation technology, in terms of utilization, PV seems to be an under-used technology.

Regarding these issues, this paper presents an experimental characterization of a methodology that maximizes grid-tied PV system operation under low-solar-radiation conditions. These new techniques could help mitigate intermittency, contribute to self-consumption [15], deploy distributed hybrid generation systems, and incorporate storage technologies over existing PV systems in an easy way, taking advantage of the electrical PV module characteristics.

From a topological point of view, different solutions have been addressed for this matter [16,17], including the incorporation of multiple maximum power-point tracking (MPPT) systems. These subsystems based on DC/DC converters generate a common DC bus that feeds a power inverter stage. In general, in commercial inverters, this stage is integrated at the factory. The proposed method allows its integration into existing conventional GCPVSs with minimal intervention on the system through a parallel electrical interconnection. Likewise, the proposed method can be extended to other photovoltaic applications through the implementation of suitably designed current control schemes, for example, in large photovoltaic systems used for water pumping where intermittency in generation capacity can cause inconveniences in operation. They are also useful when it is necessary to dynamically increase generation and or storage capacity in mini-grids that incorporate conventional photovoltaic generation.

## 2. Principle of Operation

Standard grid-connected PV systems are composed of a PV array, grid-tied inverter, and grid, as shown in Figure 1. For efficient operation, the GCPVS must inject the all-electric energy available in PV array terminals. That depends directly on the irradiance and temperature condition to which it is exposed.

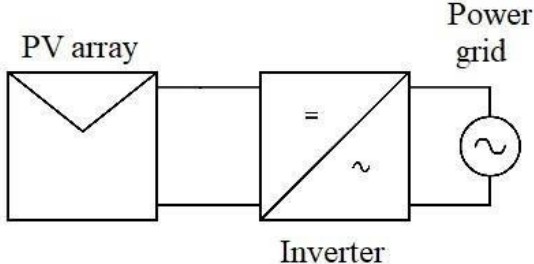

**Figure 1.** Grid-connected PV system block diagram.

A simple and accepted way to explain the PV array operation principle is through the electrical-cell-equivalent-based, five-parameter mathematical model, presented in Figure 2 [15,18]. The constant-current source ($I_{PV}$) represents photo-generated current and appears only in the presence of solar irradiation. The series resistor ($R_S$) represents the cell's contact and internal resistance, and the parallel resistor ($R_P$) has its origin in the cell's imperfections [19–21]. A PV array can be represented as an electrical series and parallel

interconnection of cells whose equivalent model can be represented by the interconnection of circuits like that presented in Figure 2.

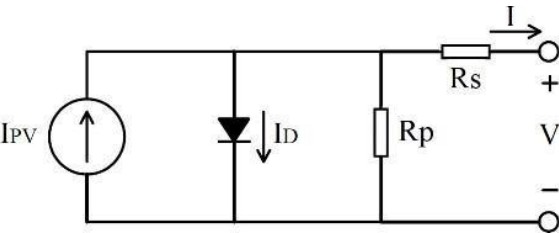

**Figure 2.** Electrical equivalent circuit for five-parameter mathematical model.

The value of the $I_{PV}$ depends mainly on the incident irradiance and is equal to zero in dark conditions (no irradiation exposure).

Connecting an external constant-current source to a dark PV cell, as shown in Figure 3, may produce similar behavior to that which it has under normal operating conditions with the incidence of irradiance. The principal difference will lie in its relative location concerning the series resistance.

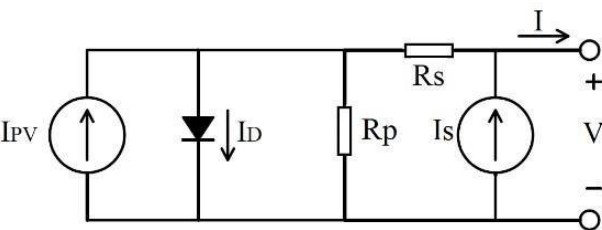

**Figure 3.** External constant-current source connection to PV cell (Dark condition requires $I_{PV}$ = 0).

These phenomena repeat if a constant-current source is connected to a dark PV array with the consequent variation in array open-circuit voltage (caused by the voltage developed in $R_S$).

If the PV array is a part of a GCPVS, this will cause the inverter to connect to the grid and inject energy coming from the constant-current source. Figure 4 shows the GCPVS diagram with a constant-current source. An isolated grid source could be another generation and/or storage technology.

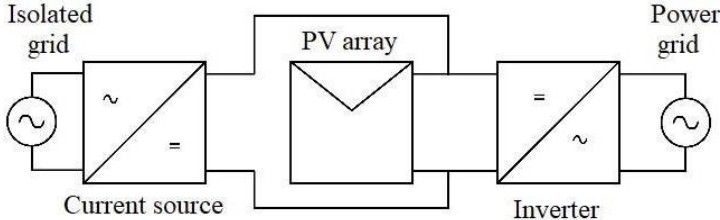

**Figure 4.** System connection diagram with a constant-current source.

Another way to represent this concept is through dark and illuminated PV array current vs. voltage (I-V) curve analysis, as it is shown in Figure 5. In dark conditions, the I-V curve is similar to a conventional silicon diode array curve [22]. With irradiation, the curve displaces to the fourth quadrant due to photo-generated current ($I_{PV}$). If an external constant-current source ($I_S$) emulates photo-generated current, the I-V curve displacement occurs, and the PV array has the same behavior as that for irradiated conditions.

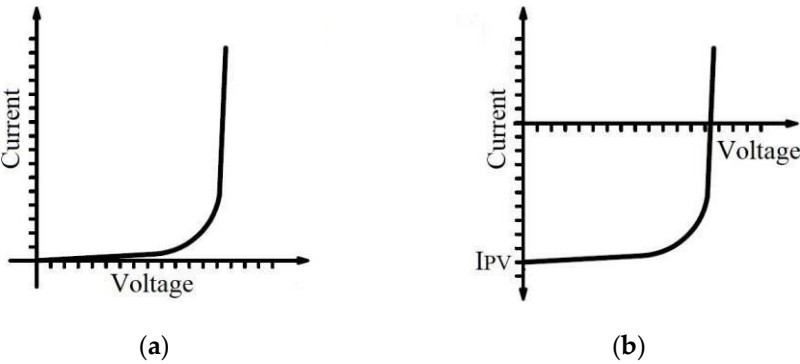

**Figure 5.** PV array current–voltage characteristics: (**a**) dark condition; (**b**) irradiated condition.

In this way, three possible scenarios could be presented:

(1) Only solar radiation produces photo-generated current ($I_P \neq 0$) on the PV array (the same case presented in Figure 5b).

(2) Solar radiation produces photo-generated current ($I_{PV} \neq 0$) and the secondary generator injects constant current to the array too ($I_S \neq 0$). The new PV array short-circuit current ($I_{SC}$) is equal to the sum of $I_{PV}$ and $I_S$ (Figure 6a).

(3) No solar radiation over the PV array ($I_{PV} = 0$), with only the secondary generator injecting the constant current, $I_S \neq 0$ (Figure 6b).

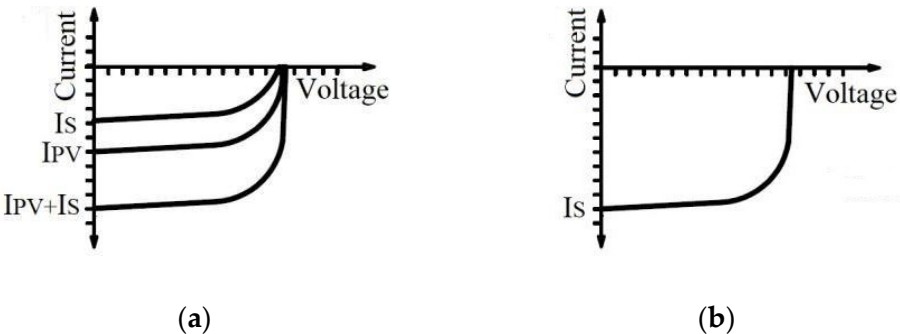

**Figure 6.** Graphic I-V curve composition for PV array and constant-current source coupling. (**a**) With solar radiation over PV array surface; (**b**) without irradiation.

The main novelty of the proposed method is that the secondary generation source can be incorporated in parallel to an operational conventional GCPVS through an efficient V/I converter, without the need to add additional stages between the PV generator and the inverter. This interconnection stage, which can be implemented through an AC/DC converter in cascade with a DC/DC converter, must have the ability to control the injected current so as not to exceed the design limits of the GCPVS. On the other hand, through external energization of the photovoltaic generator, it is possible to take advantage of the characteristics of the inverter to inject energy from another generation and/or storage source into the grid. This hybridization mechanism makes it possible to take advantage of periods of non-operation of the system and complement generation in the face of variations in the solar resource, allowing action on the intermittency inherent in the GCPVS. In this way, it is possible to improve the impact produced by the insertion of energy in the GCPVS on the quality of the electrical service in distribution systems.

The next section presents a DC/DC converter current control-based development used in an experimental setup for connecting a PV inverter to a grid with and without radiation over the PV module surface, together with an external power source (an isolated conventional grid was used).

### 3. Tension-to-Current Source DC/DC Converter Development

A 450 W buck tension-to-current source DC/DC converter was designed and implemented with 300 V (±20%) and 2 A maximum input voltage and current capabilities. Figure 7 presents a block diagram for the selected topology. The converter was based on IC TL494 [23,24], with a 33 kHz commutation frequency. The input and output filters were designed following [25].

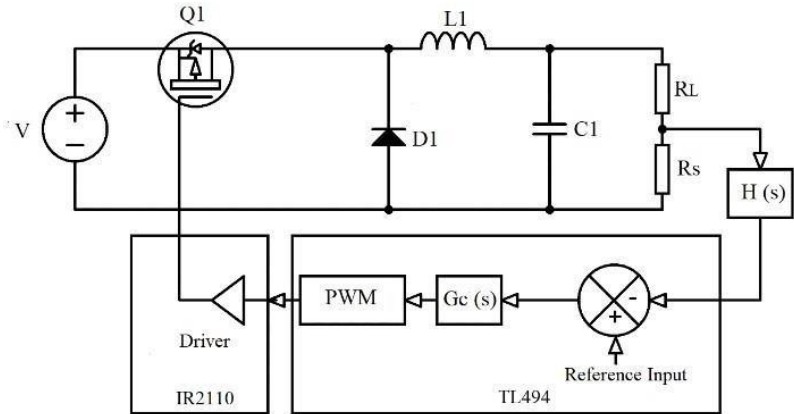

**Figure 7.** DC/DC converter block diagram.

In Figure 7, a 0.1 Ω resistor (Rs) provides a voltage as a proportion of load current for the TL494 feedback parameter. With this modification, the converter provides output constant current over a 180–230 V output voltage range. A P-channel mosfet ($Q_1$) was used in the power switch stage driven by the driver IR2110 and the TL494 output.

Table 1 summarizes the converter capabilities and the repository (https://doi.org/10.5281/zenodo.6799976) shows the schematic design of the DC/DC converter. To provide AC input capabilities to the converter, an AC/DC rectifier bridge-filter-based converter with a soft-start circuit was provided.

**Table 1.** Converter capabilities.

| Parameter | Value |
|:---:|:---:|
| Vo | 180–230 V |
| Vi | 300 V ± 20% |
| Io | 2.5 A |
| Ii | 2 A |
| Fsw | 33 kHz |
| Vripple | 0.1% Vo |
| Iripple | 6% Io |

### 4. Experimental Setup

The integration technique presented before was experimentally characterized in three ways. First, a DC/DC conversion efficiency curve was acquired for different charge conditions (different values of $I_S$). Second, global efficiency was evaluated considering absorbed energy at the DC/DC converter input vs. the active energy delivered to the AC grid with a PV inverter in a GCPVS (SIRIO 1500 W inverter) for two scenarios, with and without solar irradiance at the PV generator (illuminated and dark conditions). Third, the quality of the energy delivered to the grid was characterized with and without the DC/DC converter connection. Also, to take reference of the experimental results, the Sirio 1500 conversion efficiency was evaluated at GCPVS normal operation on a sunny day. Figure 8 shows schematics for the experimental setups.

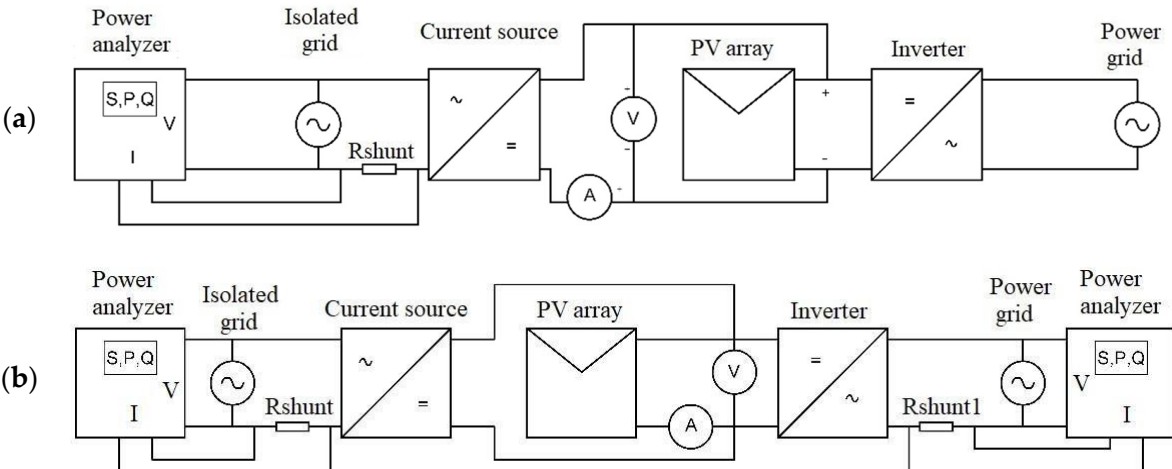

**Figure 8.** Schematics for experimental setup. (**a**) DC/DC conversion efficiency evaluation. (**b**) Characterization of inverter conversion efficiency, global efficiency evaluation, and energy delivered to AC grid.

In all cases, the inverter was connected to a series arrangement of 240 Wp PV modules (Sunmodule SW240 Poly), and the adjusted conversion efficiency function [26] was obtained and compared.

Both the DC/DC converter and the inverter for grid connection were developed, and the system, with them operating together, was experimentally characterized based on the efficiencies involved in different operating conditions. The experimental methodology proposed for this is similar to that presented by [27–29].

*4.1. AC and DC Measurement*

For AC measurements, power analyzers were used with 3 s integration time, linear averaging, and a 5.5 kHz cut-off frequency low-pass filter. DC measurements were performed with high-accuracy voltmeters and ammeters. A class 0.5 shunt was used to measure AC and DC current. Table 2 synthesizes the instrument's accuracy.

**Table 2.** Instrument accuracy.

| Instruments | |
|---|---|
| Power Analyzer—Yokogawa WT500 | |
| Power accuracy | 0.1% reading + 0.1% range |
| Frequency accuracy | 0.1 mHz |
| Agilent 34410A based Class A Power Analyzer [30] | |
| Power accuracy | 0.87% |
| Frequency accuracy | 2.5 mHz |
| FLUKE 289 and 0.5 class shunt | |
| Voltage accuracy—500 VDC | 0.03% + 2 |
| Voltage accuracy—500 mVDC | 0.025% + 2 |
| Shunt Resistor—5A/150 mV | 0.15 mΩ |

To determine the uncertainties associated with the measured magnitudes, the instrumental accuracies presented in Table 2 were used. Below, and in Sections 4.2–4.4, the equations used to determine efficiencies and the corresponding error analysis for each case are described.

Measured average power can be calculated as follows:

$$P_{avg} = \frac{1}{T} \int P_i t_i = \frac{1}{T} \sum E_i \tag{1}$$

where $P_{avg}$ is average power, $P_i$ is active power measured at instant of time $i$, $t_i$ is integration time (3 s), $T$ is measurement period (10 min), and $E_i$ is energy measured at instant of time $i$.

Error in time, defined in the instrumentation by zero-crossing time, is related to frequency accuracy or period error:

$$\Delta t = \frac{1}{f_i^2}\Delta f \tag{2}$$

where $\Delta t$ is time error, $f_i$ is measured frequency, and $\Delta f$ is frequency accuracy.

In this way, the error in power measurement can be expressed as:

$$\Delta P_{avg} = \left|\frac{\sum E_i}{T^2}\Delta T_{inst}\right| + |\Delta P_{inst}| + \left|\frac{\Delta t_{inst}\sum P_i}{T}\right| \tag{3}$$

where $\Delta P_{avg}$ is average power error, $\Delta T_{inst}$ is period error, $\Delta P_{inst}$ is active power error, and $\Delta t_{inst}$ is integration time error.

### 4.2. Inverter Characterization

The experimental setup was the same as shown in Figure 8b, without the DC/DC converter connection. Input, output power, and output power factor were acquired using two Fluke 289 multimeters at the inverter DC input and the Yokogawa WT500 at the inverter AC output. The inverter was connected to a series arrangement of seven 240 Wp PV modules that were exposed to real sun during a sunny day (Corrientes, Argentina, 10 October 2022 from 5:50 to 14:25 h). To achieve the conversion efficiency curve, a measurement was taken every five minutes.

Conversion efficiency was calculated with Equations (4) and (5).

$$\eta_{inv} = \frac{P_{AC-Inv}}{P_{DC-Inv}} \tag{4}$$

$$\Delta\eta_{inv} = \left|\frac{1}{P_{DC-Inv}}\right|\Delta P_{AC-Inv} + \left|\frac{P_{AC-Inv}}{P_{DC-Inv}^2}\right|\Delta P_{DC-Inv} \tag{5}$$

where $\eta_{inv}$ is inverter efficiency, $P_{AC-Inv}$ is inverter output power, $P_{DC-inv}$ is inverter input power, $\Delta\eta_{inv}$ is inverter efficiency error, $\Delta P_{AC-Inv}$ is inverter output power error, and $\Delta P_{DC-Inv}$ is inverter input power error.

### 4.3. DC/DC Converter Efficiency

For DC/DC converter efficiency evaluation, two experimental scenarios were provided. In the first place, six 240 Wp PV modules were connected in series at the Sirio 1500 input, and in the second place, seven PV modules were connected in order to provide different maximum power-point voltage values during tests. Figure 8a shows the experimental setup adopted. As was explained before in the text, the DC/DC converter has an AC/DC converter stage (soft-start circuit–rectifier bridge–capacitive filter) that is helpful when connecting the AC sources. In this study, an AC source was used at the developed converter input (isolated grid) to evaluate overall efficiency. In this sense, the Yokogawa WT500 was connected at the converter's input and two Fluke 289s were connected at the converter output. During the tests, the PV modules remained in the dark, and the energy provided by the converter was delivered to the grid by the Sirio inverter. The converter efficiency was calculated using the following:

$$\eta_S = \frac{P_{DC-S}}{P_{AC-S}} \tag{6}$$

$$\Delta\eta_{Source} = \left|\frac{1}{P_{AC-S}}\right|\Delta P_{DC-S} + \left|\frac{P_{DC-S}}{P_{AC-S}^2}\right|\Delta P_{AC-S} \tag{7}$$

where $\eta_S$ is converter efficiency, $P_{AC-S}$ is converter input power, $P_{DC-S}$ is converter output power, $\Delta\eta_S$ is converter efficiency error, $\Delta P_{AC-S}$ is converter input power error, and $\Delta P_{DC-S}$ is converter output power error.

### 4.4. Global Efficiency at Dark and Illuminated Conditions

The experimental setup was the same as shown in Figure 8; two power analyzers were required to characterize the developed converter input power and the Sirio 1500 output power (the electrical signals are in AC at these points). Also, to acquire the DC power delivered and/or received by the PV generator, two Fluke 289s were connected. One of them measures the current delivered/received by the PV modules and the other measures the PV voltage at the Sirio 1500 input.

First, a dark PV module condition test was performed for different DC/DC current values. In this condition, the power provided by the converter flows to the PV modules and the Sirio 1500 input. For the second test, the PV modules were exposed to the sun and, in this case, power flows from the PV modules and the developed converter to the Sirio 1500 input. This test was performed for different currents (from the DC/DC converter) and irradiance values. Irradiance was acquired with a Kipp and Zonen Pyranometer CMP10. In all the cases, power values were calculated using (1).

Global efficiency and errors were calculated for the two PV module conditions: Equations (8) and (9) for dark conditions and Equations (10) and (11) for illuminated conditions.

$$\eta_{Global} = \frac{P_{AC-Inv}}{P_{AC-S}} \tag{8}$$

$$\Delta\eta_{Global} = \left| \frac{1}{P_{AC-S}} \right| \Delta P_{AC-Inv} + \left| \frac{P_{AC-Inv}}{P_{AC-S}^2} \right| \Delta P_{AC-S} \tag{9}$$

$$\eta_{Global} = \frac{P_{AC-Inv}}{P_{AC-S} + P_{DC-FV}} \tag{10}$$

$$\Delta\eta_{Global} = \left| \frac{1}{P_{AC-S} + P_{DC-FV}} \right| \Delta P_{AC-Inv} + \left| \frac{P_{AC-Inv}}{(P_{AC-S} + P_{DC-FV})^2} \right| (\Delta P_{AC-S} + \Delta P_{DC-FV}) \tag{11}$$

where $\eta_{Global}$ is global efficiency, $P_{AC-Inv}$ is inverter output power, $P_{AC-S}$ is converter input power, $\Delta\eta_{Global}$ is global efficiency error, $\Delta P_{AC-Inv}$ is inverter output power error, $\Delta P_{AC-S}$ is converter input power error, $P_{DC-FV}$ is PV power, and $\Delta P_{DC-FV}$ is PV power error.

Harmonic distortion characterization was performed at the Sirio 1500 output, and the Yokogawa WT500 was used for this test. The total harmonic distortion in the current (THDi) and the total harmonic distortion in the voltage (THDv) were acquired. These values were compared to that acquired when characterizing the GCPVS without developed converter connections (reference test).

## 5. Results

This section presents the experimental results acquired in the performed tests. Firstly, the Sirio 1500 characterization is shown. Secondly, the developed converter characterization is presented. Thirdly, the global topology characterization results, with and without irradiance (the dark condition), are shown, and finally, a graphic comparison between the results is presented, including total harmonic distortion.

### 5.1. Inverter Characterization Results

Figure 9 shows the conversion efficiency curve as a function of the Sirio 1500 output power. The maximum overall experimental error calculated using Equation (5) is 3%, and the adjusted conversion efficiency function [21] is:

$$\eta_{inv} = \frac{P_{AC-Inv}}{5.066 + 1.061 P_{AC-Inv} - 1.799 \times 10^{-6} P_{AC-Inv}^2} \tag{12}$$

where $\eta_{inv}$ is inverter efficiency, and $P_{AC-Inv}$ is inverter output power.

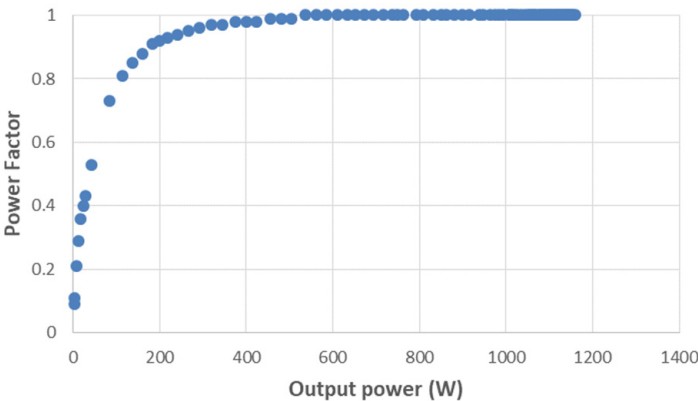

**Figure 9.** Inverter conversion efficiency vs. output power curve.

Figure 10 shows the Sirio 1500 power factor vs. output power curve. These results are useful to characterize the energy delivered to the grid as a function of the inverter operating point.

**Figure 10.** Inverter output power factor vs. output power curve.

With almost over 400 W of inverter output, the delivered energy power factor is around unity. Under that output power value, reactive energy is delivered to the grid by the Sirio 1500. This is a usual response for low-power inverters, as can be seen in [21,23].

*5.2. DC/DC Converter Characterization Results*

To acquire an efficiency curve at different operating conditions, a DC/DC converter characterization was performed in two different experimental setups, one with seven PV modules connected in series and another with six. In both scenarios, efficiency vs. output power was acquired (Figures 11 and 12). Applying Equation (7), the maximum error in determining the efficiency of the developed source was 2%.

As can be seen in Figures 11 and 12, the efficiencies of the DC/DC converter for the two different operating conditions are similar. Achieving high levels of efficiency, such as those presented in this work, is common in DC/DC BUCK topologies [24,25].

Applying the system losses model [21] and using the MATLAB curve fitting tool, the efficiency of the source for an array of six PV modules was determined as a function of the output power:

$$\eta_{Source} = \frac{P_{DC-S}}{6.728 + 0.9907 P_{DC-S} + 6.265 \times 10^{-5} P_{DC-S}{}^2} \tag{13}$$

where $\eta_S$ is converter efficiency and $P_{DC-S}$ is converter output power.

The efficiency as a function of the output power for an array of seven PV modules connected in series was:

$$\eta_{Source} = \frac{P_{DC-S}}{4.776 + 1.011P_{DC-S} + 2.863 \times 10^{-5}P_{DC-S}{}^2} \tag{14}$$

By calculating Equations (13) and (14) for an output power range from 100 to 1000 W, the efficiency curves for the two test conditions were obtained (Figure 13).

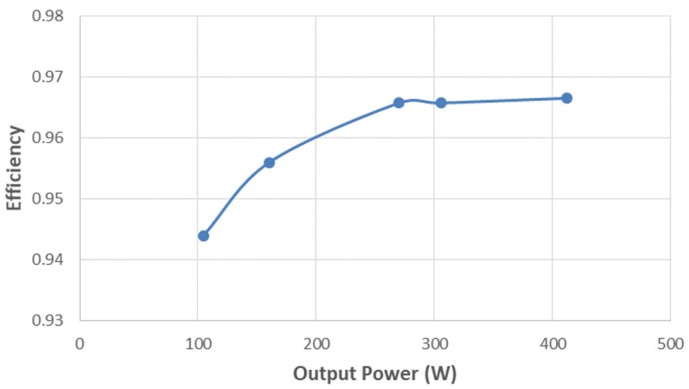

**Figure 11.** DC/DC converter efficiency vs. output power curve for six PV modules connected in series.

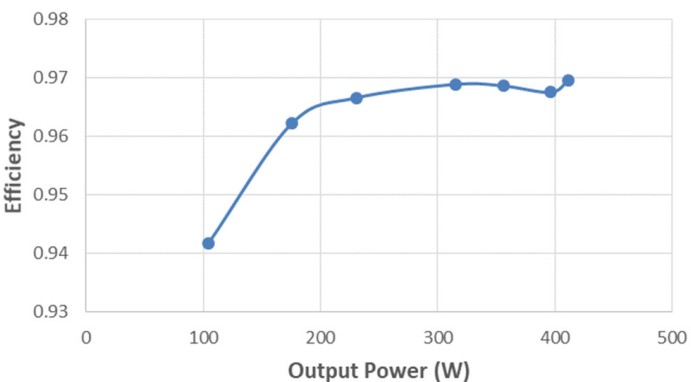

**Figure 12.** DC/DC converter efficiency vs. output power curve for seven PV modules connected in series.

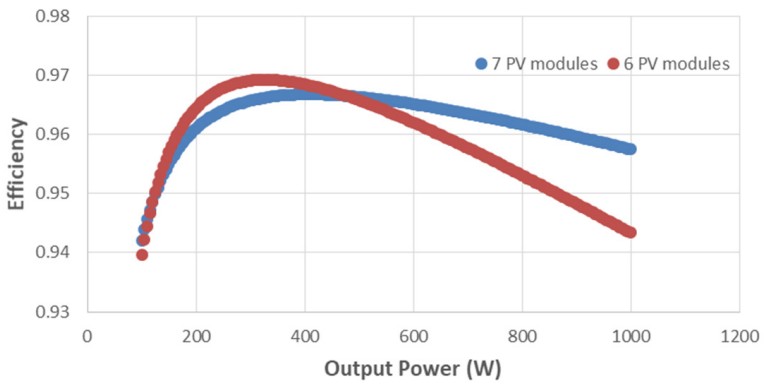

**Figure 13.** DC/DC converter efficiency vs. output power adjusted curve for seven and six PV modules connected in series.

The curves only represent approximations. In both cases, current source efficiencies greater than 90% were obtained from 100 W output. Although these efficiency values can be increased through the incorporation of more efficient electronic devices, they are considered acceptable for the intended purpose [31].

### 5.3. Global Efficiency without PV Array Solar Exposure

Global efficiency for the system in the dark condition was obtained as a function of the inverter output power (Figure 14) with a maximum error of 2%.

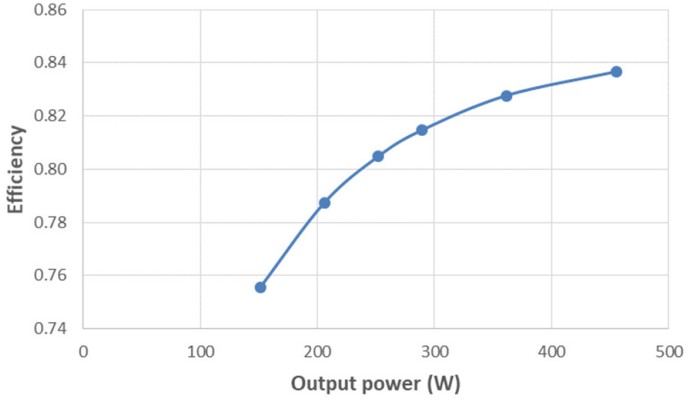

**Figure 14.** Global efficiency of the system under dark condition.

Applying the system losses model [19] and using the MATLAB R2022a curve fitting tool (cftool), the global efficiency was determined as a function of the output power.

$$\eta_{Global} = \frac{P_{AC-Inv}}{21.63 + 1.133 P_{AC-Inv} + 1.159 \times 10^{-5} P_{AC-Inv}^2} \tag{15}$$

Considering the power at the PV array, a curve of dissipated power in the PV modules was obtained as a function of the active power supplied to the developed current source (Figure 15). The maximum error in dissipated power was around 1%.

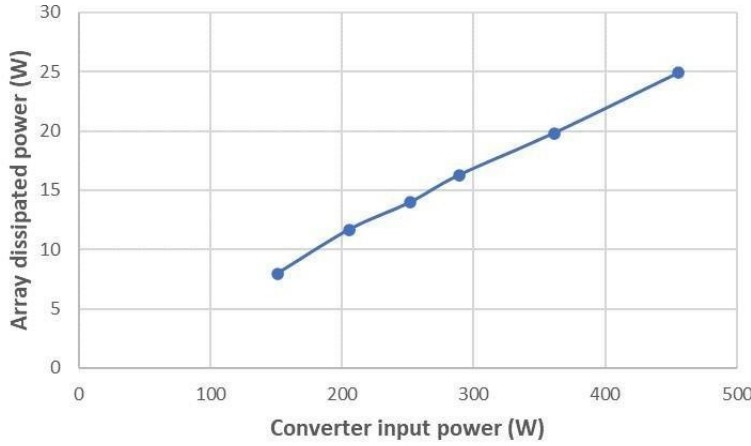

**Figure 15.** Array-dissipated power.

A linear response is observed with a slope such that the panel consumption represents approximately 5.5% of the power delivered to the current source.

Assuming a constant fill factor (*FF*) in the panel arrangement and $V_{MPP}/V_{OC} = K_1$ the current difference $\Delta I$ between the $I_{SC}$ and the $I_{MPP}$ is proportional to the *FF*, and hence the linearity observed in Figure 15 is justified, as shown in (13).

$$P_D = (I_{SC} - I_{MPP})V_{MPP} = \left(I_{SC} - \frac{FF}{K_1}I_{SC}\right)V_{MPP} = \left(1 - \frac{FF}{K_1}\right)V_{MPP}I_{SC} = (1 - \frac{FF}{K_1})P_{AC-S} \quad (16)$$

$$FF = \frac{V_{MPP}I_{MPP}}{V_{OC}I_{SC}} \quad (17)$$

where $P_D$ is array-dissipated power, $P_{AC-S} = V_{MPP}I_{SC}$ is converter input power, and $K_1$ is a constant.

### 5.4. Global Efficiency with Radiation Incidence over PV Array Surface

In this case, the global efficiency of the system under the illuminated condition (Figure 16) with a maximum error of 2.7% is presented.

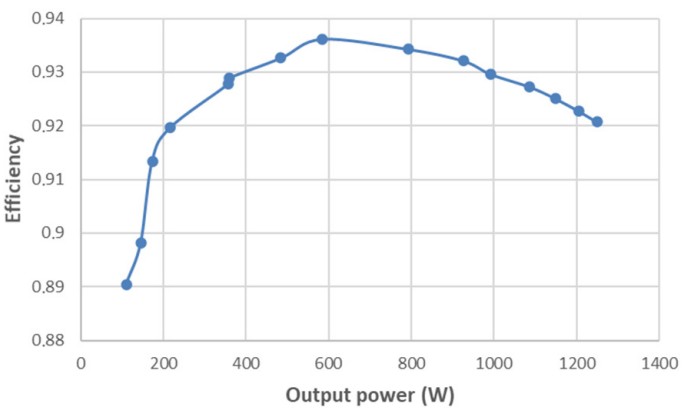

**Figure 16.** Global efficiency of the system with irradiance.

In (15), the adjusted efficiency is shown with an $R^2 = 0.99$.

$$\eta_{Global} = \frac{70.18 + 0.9817P_{AC-Inv} - 3.977 \times 10^{-5}P_{AC-Inv}^2}{89.66 + P_{AC-Inv}} \quad (18)$$

It is important to note that with irradiance and from 150 W of inverter output power, the system's overall efficiency is above 90%. It was possible to connect a second-generation system to a GCPVS working at 15% of nominal inverter power without substantially affecting its performance.

On the other hand, in this work, the current source allows manual control of the output current, but it is evident at this point that, with the help of an appropriate electronic power converter with current control capabilities, the $I_S$ from the constant-current source can be regulated to compensate the $I_{PV}$ fluctuations due to the diary radiation profile. The proposed architecture could mitigate the effects of intermittency.

### 5.5. Comparison of Results

This section presents a graphic comparison between the results obtained with the photovoltaic generator exposed to solar radiation and in dark conditions. Figure 17 presents the adjusted global efficiency with and without irradiance using (12) and (15). Figure 18 show the difference.

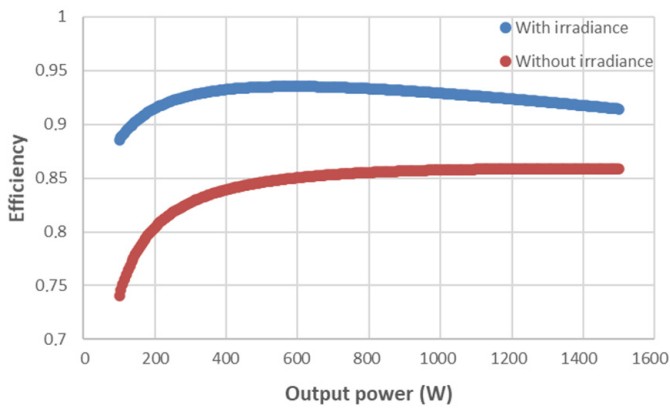

**Figure 17.** Adjusted global efficiency with and without irradiance.

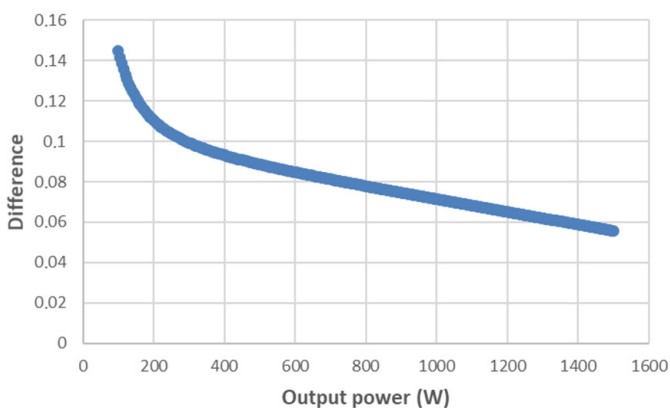

**Figure 18.** Adjusted global efficiency difference.

It was observed that from 300 W output power, the global efficiency difference is less than 10, and the trend for higher output power is 5.5%, which is the percentage of power consumed by the panels with zero irradiance.

Finally, Figure 19 presents the total harmonic distortion with and without the current source (with radiation incidence in both cases). A maximum difference in THDi of 1.3% and a maximum difference in THDv of 0.3% were observed.

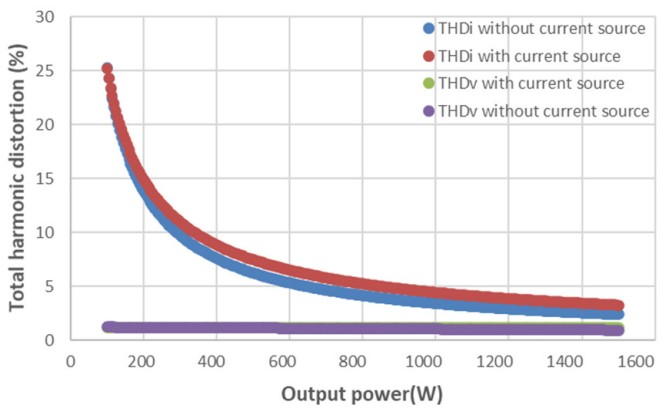

**Figure 19.** Total harmonic distortion with and without current source.

## 6. Conclusions

- An experimental characterization of a methodology for integrating difference sources and/or storage to a conventional grid-connected PV system was performed.

- The main novelty of the proposed method is that it allows taking advantage of the electrical characteristics of the photovoltaic modules to interconnect another energy supply source in parallel with a GCPVS, without further intervention. In this way, the grid-connected inverter injects energy regardless of whether solar radiation hits the photovoltaic generator.
- A DC/DC converter for constant-current source application was developed for the characterization of the current injection into a GCPVS.
- High overall yields were obtained experimentally, showing that the total harmonic distortion with the current source connected is at the same level as the GCPVS without the current source connected.
- These results show a preliminary validation of the proposed method of coupling a secondary generation system to a GCPVS.
- The presented topology opens a wide field of research in the photovoltaic systems area.

**Author Contributions:** Conceptualization, M.C. and A.F.; methodology, A.R.G.M.; validation, M.C., A.F. and L.V.; formal analysis, L.V. and J.d.l.C.H.; investigation, M.C. and A.R.G.M.; writing—original draft preparation, M.C. and A.R.G.M.; supervision, M.C.; funding acquisition, J.d.l.C.H. All authors have read and agreed to the published version of the manuscript.

**Funding:** This research received no external funding.

**Institutional Review Board Statement:** Not applicable.

**Informed Consent Statement:** Not applicable.

**Data Availability Statement:** The data will be available on a reasonable request.

**Conflicts of Interest:** The authors declare no conflict of interest.

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
