# Peer review of "New Upgrade to Improve Operation of Conventional Grid-Connected Photovoltaic Systems"

_technologies, doi:10.3390/technologies12050061_

Round 1

Reviewer 1 Report

Comments and Suggestions for Authors

This work shows a methodology for the integration of various sources and storage to a conventional grid-connected photovoltaic plant. The topic is of interest to the photovoltaic community. Furthermore, the manuscript is well organized and the method is clearly described. The experimental implementation is concise and the results obtained are good. I have just two comments ro authors :

-         Figure 6 b. Delete “Current” on the right part of the Picture.

Conclusions must be improved showing the novelty of the proposed method

Comments on the Quality of English Language

Some paragraphs need to be improved

Author Response

Thank you very much for your work and your valuable comments. We have replaid to your suggestions in an attached file.

Best regards,

Juan de la Casa

Full Professor - Universidad de Jaén (Spain)

Reviewer 2 Report

Comments and Suggestions for Authors

Overall manuscript

The authors present an interesting approach for improving the operation of GCPV systems. Below are the peer-review comments for the submitted manuscript:

Recommendation: Accept with minor revisions

Weaknesses of the manuscript:

Content:

1. Kindly provide recommendations for future research based on the findings of this study.

Referencing:

1. Kindly provide references for all equations which are not derived by the authors.

2. The references are mainly not up to date. Kindly ensure to cite lately published work related to this topic to ensure that the manuscript is up to recent knowledge.

Suggestion(s):

1. To include photographs from the preparation stage.

2. To describe the methodology using a flowchart.

Abstract:

1. The abstract does not provide key conclusions from the study. Briefly provide key conclusions from the study.

2. The abstract should consist of a brief problem statement to establish the motivation for the study and its value to the literature.

Introduction/Literature review:

1. The literature review is interesting, but not sufficient. Please add more references of recent studies covering this topic to provide further perspective on this topic.

Methods:

1. Kindly add a paragraph describing the uncertainty of the experiment.

2. Kindly provide further details into the uncertainty of the experiment (errors). For instance, state the uncertainty of the individual sensors used, and the total uncertainty of the experiment.

3. Was data collection done using a data acquisition system? If so, kindly mention it.

4. Kindly briefly discuss the main differences between the proposed methodology and other methodologies in the literature.

Results:

1. The authors provide a comparison of their findings with the literature.

2. The results (trends) should be generally compared with results of similar studies.

3. The authors describe the results; however, a discussion is lacking. Kindly provide a deeper perspective aside from describing the findings in the figures.

Conclusion:

1. Kindly ensure to provide the key conclusions in bullet points or a numbered list.

2. The conclusions should be specific and based on the outcomes of the study.

Author Response

Thank you very much for your work and your valuable comments. We have replied to your suggestions in the attached file.

Best regards,

Juan de la Casa

Full Professor - Universidad de Jaén (Spain)
